# Chance promoter activities illuminate the origins of eukaryotic intergenic transcriptions

Haiqing Xu [1,2], Chuan Li [1,3], Chuan Xu [1,4] & Jianzhi Zhang [1] ✉

It is debated whether the pervasive intergenic transcription from eukaryotic genomes has functional significance or simply reflects the promiscuity of RNA polymerases. We approach this question by comparing chance promoter activities with the expression levels of intergenic regions in the model eukaryote *Saccharomyces cerevisiae*. We build a library of over $10^5$ strains, each carrying a 120-nucleotide, chromosomally integrated, completely random sequence driving the potential transcription of a barcode. Quantifying the RNA concentration of each barcode in two environments reveals that 41–63% of random sequences have significant, albeit usually low, promoter activities. Therefore, even in eukaryotes, where the presence of chromatin is thought to repress transcription, chance transcription is prevalent. We find that only 1–5% of yeast intergenic transcriptions are unattributable to chance promoter activities or neighboring gene expressions, and these transcriptions exhibit higher-than-expected environment-specificity. These findings suggest that only a minute fraction of intergenic transcription is functional in yeast.

Many eukaryotes show pervasive transcriptions of intergenic regions[1–4]. For example, although protein-coding regions make up only ~1.5% of the human genome and genic regions constitute about one-third of the genome, RNA transcripts are detected from >75% of the genome[4]. The biological significance of intergenic transcription, however, remains controversial[5,6]. The functional hypothesis asserts that intergenic transcripts largely result from the transcriptions of uncharacterized protein-coding genes or functional noncoding genes[3,7,8]. Indeed, some long intergenic noncoding RNAs (lincRNAs) are functional[9], although the functionality of the vast majority of annotated lincRNAs is unclear[10]. Furthermore, it has been suggested that, even if the transcript of a noncoding gene is functionless, the act of transcription may regulate the expressions of other genes[11–13]. By contrast, the nonfunctional hypothesis[14,15] posits that most intergenic transcripts excluding those resulting from the expressions of neighboring genes[16,17] are nonfunctional products of chance promoter activities of intergenic sequences[18]. It is notable that, in the prokaryotic model organism *Escherichia coli*, ~10% of random sequences of ~100

nucleotides possess promoter activities comparable to that of a functional promoter and another ~60% can become such a promoter with just one mutation[19]. In the eukaryotic model organism *Saccharomyces cerevisiae*, noncoding RNAs can arise from nucleosome-depleted genomic regions by the promiscuous binding of RNA polymerase II[16–18,20], but neither the probability with which a random intergenic sequence drives transcription nor the resulting transcriptional level is known, especially in the presence of chromatin that substantially represses transcription[21]. Most lincRNAs are not evolutionarily conserved[22,23], which could mean a lack of function[6] or a rapid turnover of lineage/species-specific function[22]. Intraspecific analysis yielded mixed results on the selective constraints on lincRNAs[24].

In this study, we test the nonfunctional hypothesis by characterizing the frequency distribution of promoter activities of 120-nucleotide random sequences in yeast and comparing it with the frequency distribution of yeast intergenic expressions; the functionality of intergenic expression is invoked if the expression exceeds the chance expectation. Although the median length of yeast promoters is

[1]Department of Ecology and Evolutionary Biology, University of Michigan, Ann Arbor, MI, USA. [2]Present address: Department of Biology, Stanford University, Stanford, CA, USA. [3]Present address: Microsoft, Redmond, WA, USA. [4]Present address: Bio-X Institutes, Shanghai Jiao Tong University, Shanghai, China. ✉e-mail: jianzhi@umich.edu

455 nucleotides[25], a promoter as short as 69 nucleotides can be twice as strong as the *CYC1* promoter[26] and a 116-nucleotide synthetic promoter is similarly strong as the *TDH3* promoter[27]. *CYC1* and *TDH3* are among 30% and 0.1% of the most highly expressed yeast genes, respectively. Hence, sequences of 120 nucleotides have ample opportunities to possess promoter activities yet are not too long to lower the experimental efficiency or increase the length variation of the synthesized oligonucleotides (because of the relatively high rates of insertion/deletion errors in oligonucleotide synthesis).

Our experiment differs from past studies of eukaryotic mutant promoters in both design and purpose. We investigate the promoter activities of completely random sequences, while past studies examined activities of promoters that were created by mutating a native promoter[28,29] or were built on a core promoter scaffold[27,30–33]. We aim to estimate the probability distribution of promoter activities of random sequences, while past studies aimed to identify crucial elements of a particular functional promoter or sequence features of active promoters with a canonical scaffold.

## Results

### Estimating the promoter activities of random sequences
We began by synthesizing oligonucleotides each comprising a 120-nucleotide completely random sequence (the random promoter) and a 20-nucleotide completely random sequence (the barcode) interleaved with invariant sequences (primer sites) (Figs. 1a, S1a). Because the genomic location has a much smaller influence than the promoter strength on the gene expression level[34], we integrated the above oligonucleotides to an intergenic position in the yeast genome that permits sensitive quantification of promoter activities (Fig. S2, Table S1), using CRISPR/Cas9 in a large-scale transformation (see Methods). A *CYC1* terminator was placed upstream of the random promoter to minimize the influence of any upstream transcriptional activity (Fig. 1a). We respectively created a negative control where the random promoter was absent and a positive control where the random promoter was replaced with the promoter of the yeast *PSP2* gene. We constructed eight versions of each of the negative and positive controls using different barcodes to confirm the reliability of barcode expression measurements.

The library, along with the controls, was cultured in three replicates in a rich medium (YPD) or a minimal medium (SCD) to the exponential growth phase. We extracted and amplified DNAs from barcodes (Fig. S1b) and sequenced them using 150-nucleotide paired-end Illumina sequencing. From the same samples, we extracted total RNAs, reverse-transcribed mRNAs from the barcodes, and sequenced the corresponding cDNAs using the same platform. The number of cDNA reads divided by the number of DNA reads for each barcode, upon normalization, is an estimate of the barcode expression level and the activity of the corresponding random promoter (Fig. 1a). From the library, we also Illumina-sequenced the insert to determine the sequence of the random promoter linked with each barcode (Fig. 1a).

We focused on barcodes with at least 100 DNA reads in each replicate to allow relatively precise estimation of their expression levels; 49,169 and 146,291 barcodes passed this criterion in YPD and SCD, respectively. In YPD, the barcode DNA read number is highly correlated between replicates (Figs. 1b, S3a, b), while the cDNA read number is less well correlated (Figs. 1c, S3c, d), and the expression level correlation is even weaker (Figs. 1d, S3e, f). The reduced correlation in expression level is due to the existence of many lowly expressed barcodes; the correlation is substantially higher when the 1% of the barcodes with the most cDNA reads are examined (insets in Figs. 1d, S3ef). The same is true in SCD (Fig. S4a–i). When culturing the yeast library, we included high fractions of controls, resulting in a high expression correlation across replicates for the controls (Fig. S5). To verify the bulk sequencing-based promoter activity estimation, we selected several promoters with a wide range of activities but low across-

replicate variations, reconstructed them, and measured their expressions individually by reverse transcription-quantitative polymerase chain reaction (RT-qPCR; see Methods). Expression estimates from RT-qPCR agreed well with those from bulk sequencing (Figs. 1e, S4j).

### A large proportion of random sequences have promoter activities
By comparing with the negative control, we found that 63.2% and 41.4% of the random sequences have significant promoter activities in YPD (Figs. 2a, S6a, b) and SCD (Figs. 2b, S6a, b), respectively. Because our positive control−*PSP2*−may not have the same expression level in different environments, hereinafter we use the median expression level of all yeast native genes in the relevant medium as the reference (by comparing the *PSP2* expression level with the reference in published RNA-seq data; see Methods). About 0.024% and 0.029% of the random sequences have significantly higher promoter activities than the reference in YPD and SCD, respectively (Figs. 2a, b, S6c). Similar results were obtained when different cutoffs higher than 100 DNA reads per barcode were used in analyzing barcode expressions (Fig. S6). Most (90%) random promoters have activities below the 21st (or 12th) percentile of yeast native promoter activities in YPD (or SCD), while the strongest random promoter observed is comparable in activity to the 85th percentile of the native promoters in both growth conditions (Fig. 2c, d, Table S2).

### Features associated with the random promoter strength
Identifying sequence features associated with the promoter strength is important for understanding the mechanistic basis of the promoter activity and for synthetic biology[27,30–33,35]. In both YPD (Fig. 3a) and SCD (Fig. S7a), a positive correlation exists between random promoter strength and promoter GC content (see Methods). Interestingly, for yeast native promoters, such a positive correlation exists only for relatively strong promoters; for relatively weak promoters, the correlation is negative (Figs. S8a, b, S9a, b). About 20% of yeast native promoters contain TATA boxes[36]. We found that, for both random (Figs. 3b, S7b) and native (Figs. S8c, d, S9c, d) promoters, there is a positive correlation between promoter strength and TATA-box presence.

Due to their short and degenerate sequences, transcription factor (TF) binding sites (TFBSs) can easily arise in a random sequence[31]. Based on 196 known yeast TFs and their TFBSs[37], we found on average 21 forward and 24 reverse TFBSs (Fig. 3c) per random promoter (see Methods). For each TF, we statistically tested if random promoters with and without its TFBSs have significantly different activities. At the false discovery rate (FDR) of 0.05, the promoter activity is significantly influenced by the forward TFBSs of 111 TFs and reverse TFBSs of 114 TFs in YPD. The corresponding numbers are 64 and 75, respectively, in SCD. For each medium and orientation, the distribution of the $P$ values from the above tests is highly left skewed for real data but is approximately uniform upon the shuffling of promoter strengths among promoters (Figs. 3d, S10), confirming the genuine impacts of TFBSs. Previous work showed that some TFBSs have orientation-specific effects in promoters with canonical scaffolds[31,33]. We assessed the expression effect of each TF by the median activity of random promoters with corresponding TFBSs, relative to that without them. Overall, the effects of a TF in the forward and reverse orientations are only weakly positively correlated (Figs. 3e, S11), with many TFs showing orientation-specific effects and many showing orientation-independent effects (Figs. 3e, S11). Gene expression is often environment-dependent because of environment-dependent TF expressions or actions. When we focused on the 92 TFs with significant effects in both YPD and SCD (regardless of orientation), only three TFs showed opposite effects in the two media (Fig. S12), suggesting that opposite actions of the same TF between two environments are rare. Fig. S13 shows three examples of strong random promoters, with perfectly matching TFBSs indicated.

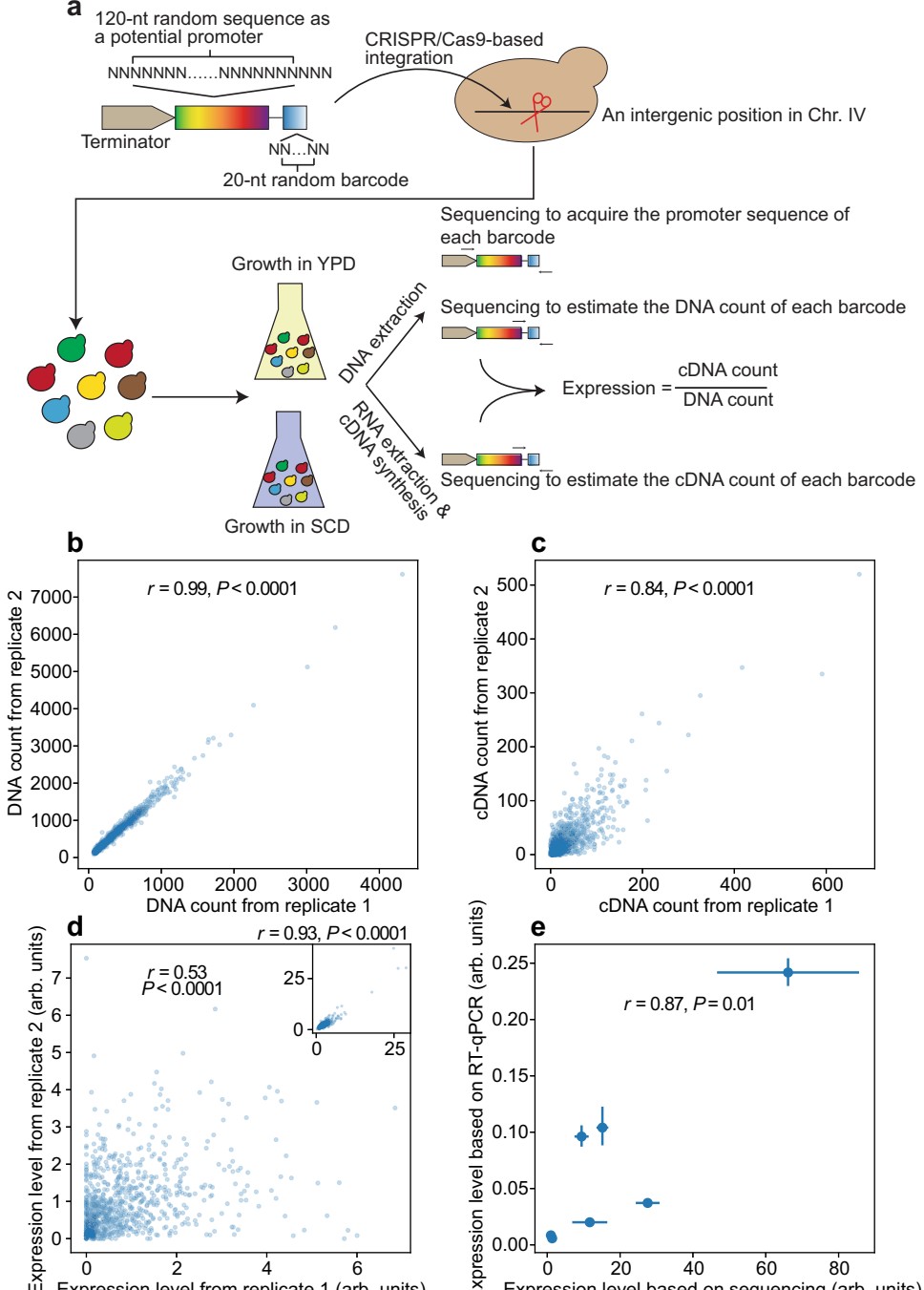

**Fig. 1 | Estimating the promoter activities of random sequences in YPD.**
**a** Experimental procedure. **b**, **c** DNA (**b**) and cDNA (**c**) counts of each barcode in two experimental replicates. For clarity, 1000 randomly sampled barcodes are shown. Pearson's correlation (*r*) and its associated *P* value based on all barcodes are presented. **d** Estimated barcode expression levels in two replicates. Expression level is measured by the cDNA count divided by the DNA count of the barcode. The same 1000 barcodes as in (**b**) and (**c**) are shown. Pearson's *r* and associated *P* value based on all barcodes are presented. The inset shows the expression levels of 1% of

genotypes with the highest cDNA counts and associated statistics. **e** Expression levels measured by bulk sequencing is strongly correlated with those measured by RT-qPCR in five reconstructed genotypes, a randomly picked negative control, and a randomly picked positive control. Mean expressions and standard errors are shown by dots and error bars, respectively. Pearson's correlation (*r*) and the associated *P* value between the two measurements are presented. All *P* values are from two-tailed tests. Source data are provided as a Source Data file.

## Most yeast intergenic expressions are explainable by chance promoter activities or neighboring gene expressions

To understand yeast intergenic expressions in the light of chance promoter activities, we first examined two existing RNA sequencing datasets[38,39] of the same yeast strain and similar growth conditions as in our random promoter experiments. To allow fair comparisons with the expressions of 20-nucleotide barcodes, we divided each genic or

intergenic region into 20-nucleotide contiguous windows, estimated the expression level of each window (Fig. S14a), and validated this measurement by benchmarking with the canonical estimates of genic expressions (Fig. S14b; see Methods). Genic as well as intergenic expressions are highly correlated across replicates (Fig. S14c, d). We subsequently generated the expression distribution of intergenic windows (Fig. 4a, b). Notably, only 0.8% and 1.3% of intergenic windows

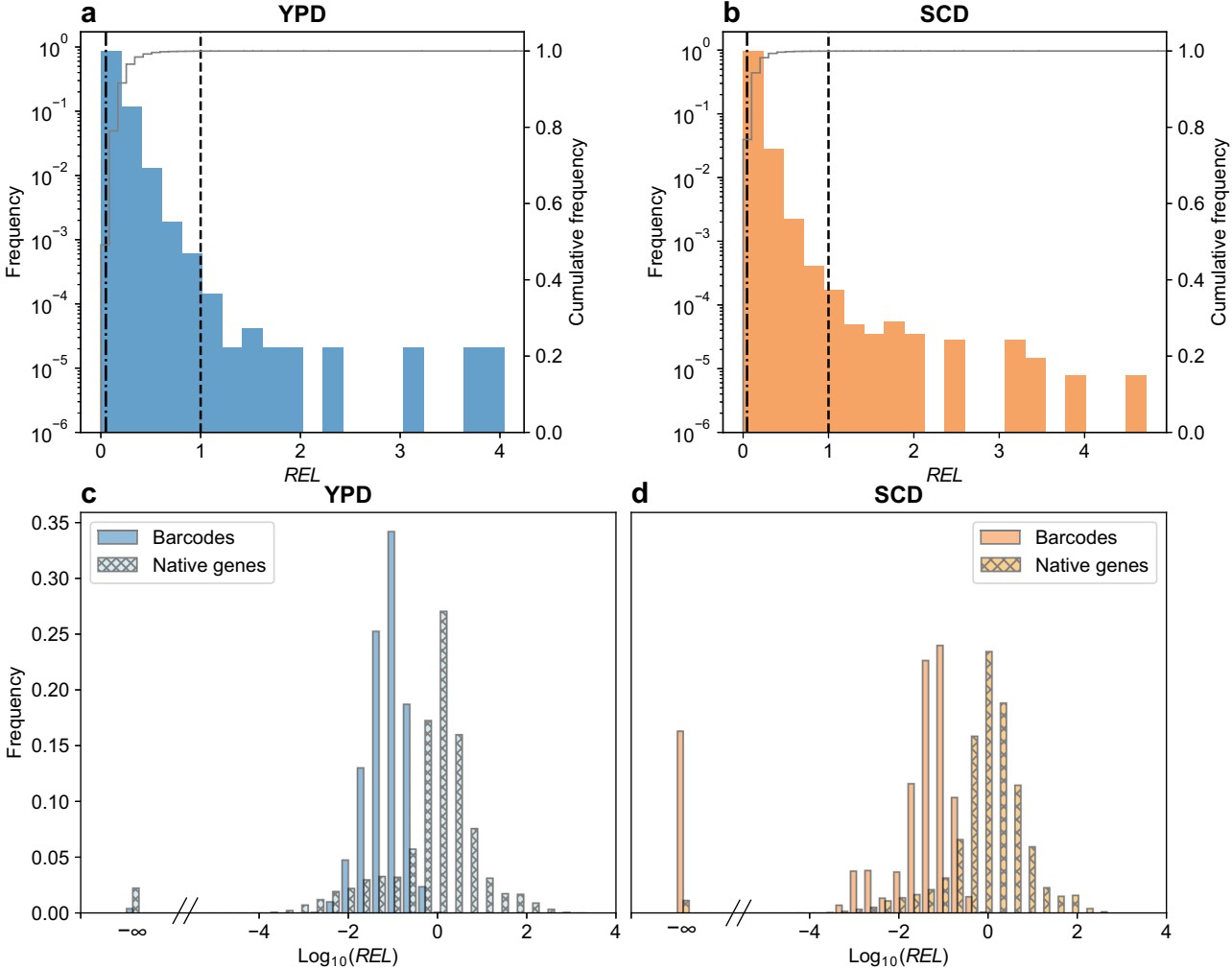

**Fig. 2 | Frequency distributions of relative expression levels (*REL*s) of barcodes and native genes. a**, **b** Frequency distribution of the barcode expression level relative to the reference (median expression level of yeast native genes) in YPD (**a**) or SCD (**b**). The left and right vertical dashed lines indicate the mean of the negative control (no promoter) and the reference, respectively. Frequency is shown by bars whereas the cumulative frequency is indicated by the gray line. Note the difference between the left and right Y-axis scales. **c**, **d** Frequency distributions of expression levels of barcodes and yeast native genes relative to the reference in YPD (**c**) or SCD (**d**). In (**c**) and (**d**), negative infinite values come from the $\log_{10}$ transformation of the expression levels for barcodes with zero cDNA counts. Source data are provided as a Source Data file.

are significantly more highly expressed than the reference in YPD and SCD, respectively.

To directly compare the expression levels of intergenic windows with those of the barcodes, we computed relative expression levels (*REL*s) by dividing the raw expression levels by the reference. We then compared the fraction of intergenic windows whose *REL*s significantly exceed various cutoffs with the corresponding fraction of barcodes. While both fractions decrease with the cutoff, the latter drops more quickly than the former and becomes smaller than the former when the *REL* cutoff is 0.3 in YPD (Fig. 4c) and 0.1 in SCD (Fig. S15a). We then computed for each expression bin the proportion of intergenic windows whose expressions can or cannot be explained by random promoter activities. In YPD, all intergenic expressions with *REL*s not significantly higher than 0.5 are explainable by chance promoter activities. Starting from the *REL* bin of 0.5–0.6 (i.e., *REL* is significantly higher than 0.5 but not significantly higher than 0.6), we observed larger and larger proportions of intergenic windows whose expressions cannot be explained by chance promoter activities (Fig. 4d). Overall, 1.7% of intergenic windows could not have their YPD expressions explained by chance promoter activities. The corresponding value is 7.6% in SCD (Fig. S15b).

Although we have minimized the potential influence of neighboring gene expressions on intergenic expressions by extending the 5′ and 3′ untranslated regions (UTRs) of neighboring genes (see Methods), it remains possible that some intergenic expressions reflect the bi-directional promoter activities or transcriptional readthroughs of neighboring genes. Indeed, we observed a significant positive correlation between the expression level of an intergenic region (i.e., the mean expression level of all windows in the intergenic region) and the mean expression level of its two neighboring genes (Fig. S16; see Methods). We progressively excluded intergenic windows with the highest neighboring gene expressions till the potential influence of neighboring gene expressions was no longer significant (Fig. S17; see Methods). Afterwards, only 1% and 5% of intergenic windows have expressions unexplained by chance promoter activities in YPD and SCD, respectively.

Another confounding factor is that we studied random promoter sequences of 120 nucleotides while an intergenic region can be longer than 120 nucleotides. More importantly, the expressions of different windows in an intergenic region are likely interdependent. To circumvent these problems, instead of using each intergenic window as a unit, we used each intergenic region as a unit and estimated that 2.9% and 4.9% of intergenic regions have expressions unexplainable

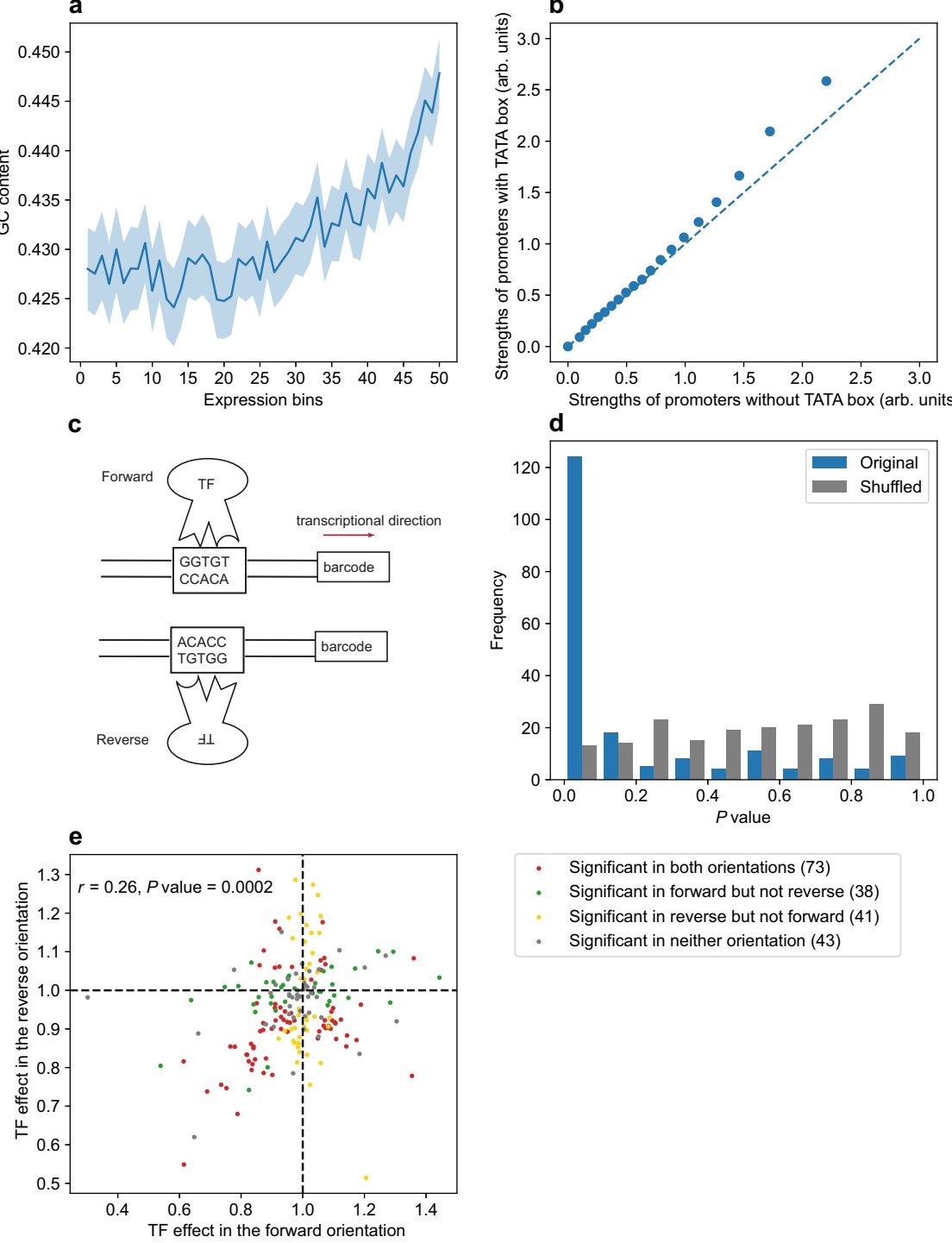

**Fig. 3 | Sequence features associated with the random promoter strength in YPD. a** The GC content in a random promoter increases with the promoter strength. The promoters are divided into 50 equal-size bins by their strengths. Shown are the mean GC content of each bin, with the shaded area indicating the 95% confidence intervals of the mean GC content. Spearman's rank correlation between the promoter strength and GC content for unbinned data is $\rho = 0.08$ ($P = 5.6 \times 10^{-65}$). **b** Quantile-quantile plot showing the probability distributions of strengths of random promoters with and without TATA boxes. The dots show the 0th, 5th, 10th, ... and 95th percentiles of the data in promoter strength. The strengths of promoters with TATA boxes are significantly higher than those without ($P = 0.01$, Wilcoxon rank-sum test). **c** Schematics illustrating the binding of a TF to a promoter in two directions (the TFBS being GGTGT). **d** Frequency distribution of

$P$ values from Wilcoxon rank-sum tests of equal activities between random promoters with binding sites of a particular TF (on the forward strand) and those without. The blue color indicates $P$ values from the original data, whereas the gray color indicates $P$ values from the data in which the promoter activity is randomly shuffled among promoters. **e** TF effects on the forward and reverse orientations. The effect of a TF in an orientation is measured by the median activity of promoters with the binding sites of the TF in the orientation concerned relative to that of promoters without the binding sites in the orientation concerned. Each dot represents a TF, and colors indicate results from Wilcoxon rank-sum tests. Number of TFs belonging to each of the four categories is shown in the parentheses. All $P$ values are from two-tailed tests. Source data are provided as a Source Data file.

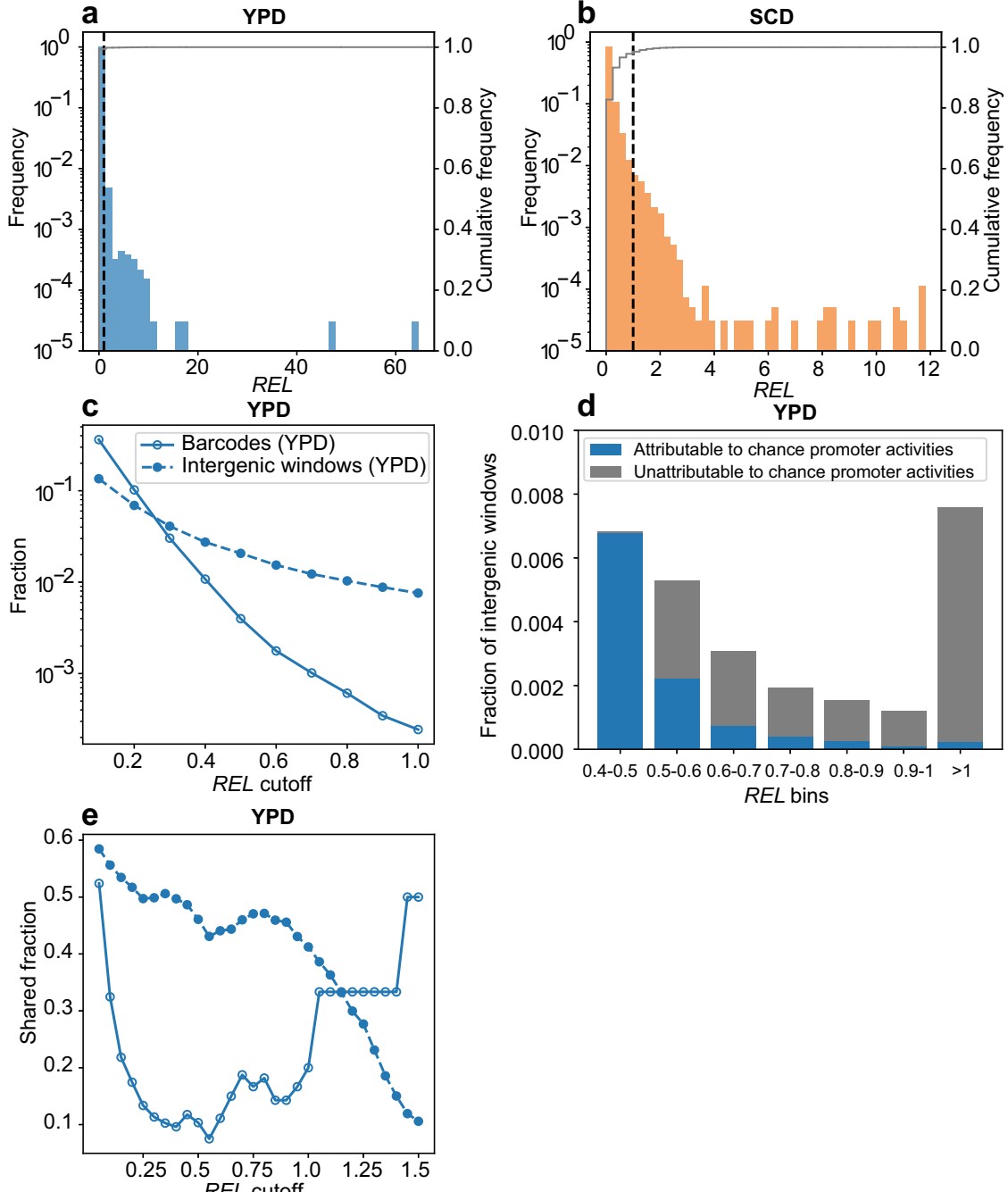

**Fig. 4 | Intergenic expressions attributable to chance promoter activities.**
**a**, **b** Frequency distribution of relative expression levels (*REL*s) of intergenic windows in YPD (**a**) or SCD (**b**). *REL* equals the expression level divided by the reference (median expression level of yeast native genes) under the same environment. Frequency is shown by bars whereas the cumulative frequency is indicated by the gray line. The vertical dashed line indicates *REL* = 1. Note the difference between the left and right Y-axis scales. **c** Fractions of barcodes or intergenic windows with *REL*s in YPD significantly higher than various cutoffs. For instance, the bin of 0.6 includes

all barcodes or intergenic windows with *REL*s significantly exceeding 0.6.
**d** Fractions of intergenic windows whose YPD expressions are attributable (blue) or unattributable (gray) to chance promoter activities. For example, the bin of 0.5–0.6 includes intergenic windows with *REL*s significantly higher than 0.5 but not significantly higher than 0.6. Only bins containing gray areas are shown. **e** Fraction of barcodes or intergenic windows with *REL*s significantly exceeding a cutoff in YPD that have *REL*s significantly exceeding the same cutoff in SCD. Symbols follow (**c**). Source data are provided as a Source Data file.

by chance promoter activities or neighboring gene expressions (Tables S3, S4; see Methods).

## Higher-than-expected environment-specificity of a minority of intergenic expressions
The activities of a promoter in different environments could be correlated. For example, 36.3% of barcodes have *REL*s significantly

exceeding 0.1 in YPD (Fig. S18a). Among barcodes with *REL*s significantly exceeding 0.1 in SCD, however, 64.8% have *REL*s significantly exceeding 0.1 in YPD (Fig. S18c), indicating nonindependent (or shared) barcode expressions in the two environments. Such nonindependence also exists for intergenic expressions (Fig. S19). If higher intergenic expressions are more likely to be functional (Fig. 4d), we might also expect them to show less sharing between environments as

a result of environment-specific demand of function. Indeed, the shared fraction of intergenic windows decreases with the *REL* cutoff, while no such trend exists for barcodes, which have no functional expressions (Figs. 4e, S15c).

## Variation among replicates

Throughout the analysis, we merged the data from the three biological replicates when measuring random promoter activities (i.e., combined analysis). To assess the variation among the replicates, we individually analyzed the data from each replicate (i.e., individual analysis). The results from the individual analysis are overall similar to those from the combined analysis (Table S5). Furthermore, the active promoters (i.e., those driving significantly higher expressions than the negative control) discovered from each replicate largely overlap with those discovered from the combined analysis (Fig. S20).

## Discussion

In summary, we found that 41–63% of 120-nucleotide random sequences have significant promoter activities in *S. cerevisiae*, demonstrating the easiness for a random sequence to be transcribed by chance even in eukaryotes. However, the probability is ~0.025% for a random promoter to be significantly stronger than the median promoter activity of yeast native genes, contrasting the observation in *E. coli* where 2.5% of random promoters are as strong as the induced *lac* promoter[19,40], which ranks in the top 3% of all *E. coli* native promoters in strength. This disparity could be due to the chromatin structure in eukaryotes[21] and/or the lack of consensus sequence in yeast that is analogous to the short motifs bound by the canonical $\sigma^{70}$-RNAP in *E. coli*[40]. Indeed, although random promoters with TATA boxes tend to be stronger than those lacking TATA boxes, a sizable fraction of the former (22.8% in YPD and 39.4% in SCD) do not have detectable activities (Fig. S21).

We investigated the relationship between various sequence features (nucleotide composition, TATA box, and TFBSs) and the promoter strength in the absence of a core promoter scaffold. Contrasting the observation in native promoters[35,41], we found the GC content to be positively correlated with the promoter activity in random sequences, suggesting that GC content and the core promoter scaffold might interact in influencing the promoter activity. About 82% of all types of TFBSs examined had a significant effect on the random promoter activity in at least one condition, suggesting that the random promoter activity may simply require the opening of the chromatin, which can be accomplished by the binding of TFBSs by TFs[42]. However, we also observed pervasive orientation-specific effects of TFBSs on the promoter activity (Figs. 3e, S11). Therefore, regulatory sequences by themselves can influence expression independently of the core promoters in both orientation-dependent and orientation-independent manners.

We found that 1–5% of yeast intergenic transcriptions, especially those exhibiting relatively high expressions, are attributable to neither chance promoter activities nor neighboring gene expressions, so are putatively functional. Consistent with this finding is the observation of a higher-than-expected environment-specificity of the relatively high intergenic expressions. Due to the drastic genome size variation across eukaryotes, it is unclear whether our findings in yeast on intergenic expression are directly applicable to other eukaryotes. But our approach is likely adaptable for studying the functional significance of intergenic transcriptions in a wide variety of eukaryotes.

## Methods
### Media used
YPD medium: 10 g/l of yeast extract, 20 g/l of peptone, and 20 g/l of glucose. YPAD medium: YPD medium plus 80 mg/l of adenine hemisulfate. SCD medium: 1.7 g/l of yeast nitrogen base (YNB), 5 g/l of ammonium sulfate, 0.79 g/l of complete supplement mixture (CSM),

and 20 g/l of glucose. SCD – Ura medium: 1.7 g/l of YNB, 5 g/l of ammonium sulfate, 0.77 g/l of CSM without uracil, and 20 g/l of glucose.

### Random promoter library construction
The 120-nucleotide random promoter and 20-nucleotide random barcode were synthesized by IDT as part of 200-nucleotide oligos. In the synthesis, equal amounts of A, T, G, and C were used for the promoter and barcode regions. We used 20-nucleotide barcodes because the large barcode space ($4^{20}$) relative to the number of barcodes in the library means that sequencing or PCR errors are extremely unlikely to convert one barcode in the library to another one in the library. The random promoter and barcode were flanked by constant regions as primer binding sites (Fig. S1a). The primer binding site downstream of the barcode has no homologous sequence in the yeast genome, ensuring that the cDNA generated is exclusively from the mRNA of the barcode.

### Genomic integration location
We compiled several RNA-seq datasets (PRJNA392312, PRJNA315924, PRJNA238899, and PRJNA239408 from NCBI) to screen for an intergenic region with a potentially high expression dynamic range. We found several candidates based on the following criteria: (1) at least 50 nucleotides long, (2) not overlapping with genes, (3) far from telomeres and centromeres, and (4) with a total of about 50–200 reads mapped to the region in 20 RNA-seq datasets in the above NCBI BioProjects. Using RT-qPCR, we quantified the expressions of the positive and negative controls (see the following section) integrated into five candidate locations (Table S1). To quantify the amount of genomic DNA contamination in the RNA sample, we performed a control experiment with no reverse transcriptase; signals in this control would arise from genomic DNA instead of RNA. Due to extremely low expressions that we are attempting to measure in this study, the above control is critical. We picked the intergenic region between *HSP31* and *FIT1* on Chr. IV as the site of integration of our library (Table S1). This site showed a relatively large difference in expression level between the positive and negative controls. Furthermore, virtually no signal was detected in the control experiment without reverse transcriptase (Fig. S2).

### Strain construction
To improve the efficiency of CRISPR/Cas9-based integration of our library of random promoters, we first used CRISPR/Cas9 to integrate a synthetic landing pad (SLP) into the aforementioned genomic integration site of the laboratory yeast strain BY4741. The SLP contains a *CYC1* terminator, used to prevent the transcription from the upstream of the integration site, and three de novo CRISPR/Cas9 targeting sites each with a 20-nucleotide Cas9 target sequence plus a three-nucleotide protospacer adjacent motif (PAM) site. Next, we integrated the random promoter library at the SLP by CRISPR/Cas9 in a large-scale liquid transformation modified from an existing protocol[43]. Specifically, we followed the same procedure of a 100× transformation until the plating step. Instead of plating cells onto the selective plate, we grew all the transformants in liquid culture for selection. We expected to obtain about 1000 transformants per 1× transformation, equivalent to 100,000 transformants in a 100× transformation ideally. In the end, with five parallel transformations, we acquired about 200,000 transformants (estimated from sequencing). Besides random promoters, we also created a positive control (*PSP2* promoter) and a negative control (no promoter) by CRISPR/Cas9. The *PSP2* promoter is one of the weakest constitutive promoters used in yeast synthetic biology[44] so is suitable for comparison with random promoters, which are expected to be weak.

Five random promoters covering a large dynamic range (~50 fold) were chosen based on their relatively high consistency in expression level across replicates. Their promoter-barcode pairs were synthesized, amplified, and integrated into SLP to create these genotypes independently.

## Large-scale liquid transformation

YPAD medium (250 ml) was inoculated by a single colony of the yeast strain BY4741 picked from a fresh YPAD plate; the yeast culture was incubated for 24 h at 200 rpm and 30 °C. About $1.25 \times 10^{10}$ cells were added to 2250 ml pre-warmed 2× YPAD medium (initial density = $5 \times 10^6$ cells/ml). The cells were allowed to grow for 4.5 h at 30 °C and 200 rpm until the density reached at least $2 \times 10^7$ cells/ml. Standard transformation steps were then performed. 0.1% of the final transformants were plated onto SCD − URA plates to estimate the transformation efficiency, and the rest of the transformants were resuspended in 2000 ml of 2× SCD − URA liquid culture and redistributed to 250 14 ml falcon tubes each containing 8 ml of culture. The falcon tubes were put on a large roller and incubated at 30 °C for 48 h. Afterwards, $1 \times 10^9$ cells were collected, washed, and resuspended into 200 ml YPD medium and cultured for 24 hrs. This step was to remove the Cas9 plasmid and to revive cells. About $5 \times 10^9$ cells were then collected and diluted in 15% glycerol to a density of $5 \times 10^7$ cell/ml and were stored at −80 °C.

## Library preparation and Illumina sequencing

The glycerol stocks of the random promoter library and controls were mixed in a 100:5:1 ratio of the random promoter library to the negative control to the positive control. The mixture was precultured in a 250 ml flask with 50 ml YPD at an initial density of $5 \times 10^6$ cells/ml for 24 h. The resulting culture was diluted in 20 ml YPD or SCD medium to an initial density of $5 \times 10^6$ cells/ml in a 100 ml flask, with three replicates per medium type. The cell cultures were always incubated at 30 °C with 250 rpm. After 8 hrs of culturing, genomic DNA was extracted from $3 \times 10^8$ cells per replicate using Masterpure™ Yeast DNA Purification Kit, whereas mRNA was extracted from $1.5 \times 10^8$ cells per replicate using RNeasy Mini Kit.

To retrieve the linkage information between promoters and barcodes, we used pairs of primers containing Illumina sequencing adapters to amplify the promoter-barcode cassette in the construction of the sequencing library. Using 200 ng of genomic DNA as templates, we conducted two parallel 18-cycle PCR reactions for each biological replicate. The resulting amplicons were combined, purified, and sequenced by 150-nucleotide paired-end Illumina sequencing (HiSeq 4000).

To generate DNA reads for the barcode region, we used a two-step PCR strategy. In the first step, primers with unique molecular identifiers (UMIs) were used to amplify the barcode region in a PCR reaction with only three cycles. The UMIs used were 6-nucleotide random sequences to mark individual DNA molecules. We conducted four parallel PCR reactions per biological replicate. The resulting PCR products for each biological replicate were purified and concentrated. In the second step, we used pairs of primers containing both the sample index and Illumina sequencing adapters to amply the previous products in an 18-cycle PCR reaction. The resulting amplicons were combined, purified, and sequenced by 150-nucleotide paired-end Illumina sequencing (HiSeq 4000).

To generate cDNA reads for the barcode region, we first reverse-transcribed the mRNA into cDNA from 2.4 μg mRNA per reaction (SuperScript® III First-Strand Synthesis System for RT-PCR). The cDNA was then amplified using the aforementioned two-step PCR strategy with UMIs. The resulting amplicons were combined, purified, and sequenced by 150-nucleotide paired-end Illumina sequencing (HiSeq 4000).

Notably, the number of genotypes obtained in YPD is only about one-third of that in SCD. This difference may be due to the higher growth rate of yeast in YPD than in SCD. Specifically, some cells may have a longer lag time by chance during the preculture stage, rendering their frequencies lower than those with a shorter lag time. This effect is intensified in YPD because of the higher growth rate in YPD than in SCD, reducing the number of genotypes obtained in YPD.

The total sequence space for a 120-nucleotide random promoter is as large as $4^{120}$. Whether our random library could accurately capture the distribution of chance promoter activities depends on whether our library is randomly distributed in the large sequence space. In theory, the expected Hamming distance between two random 120-nucleotide sequences is $120[(1 − f_A)f_A + (1 − f_T)f_T + (1 − f_G)f_G + (1 − f_C)f_C]$, where $f_X$ is the frequency of nucleotide $X$. Under equal frequencies for the four nucleotides, two random promoters should differ at 90 sites on average. However, we found that our random promoter library does not have equal frequencies for the four nucleotides, probably because of the variation introduced in the oligonucleotide synthesis. Instead, we found $f_A = 24.3\%$, $f_T = 32.4\%$, $f_G = 25.7\%$, and $f_C = 17.6\%$. So, the GC content of the random promoters in our library is 43.3%, which incidentally is closer than the GC content in our design (50%) to yeast's genomic GC content (38%). The mean Hamming distance expected from the above nucleotide frequencies is 88.78. We observed an average Hamming distance of 88.71 in our library, suggesting that the promoters in our library are randomly distributed in the large sequence space given the nucleotide frequencies.

## Influences of potential errors in library construction and sequencing

Potential PCR and sequencing errors have no impact on identifying random promoters or random barcodes because the expected sequence difference between two different barcodes or promoters (see the next section) is much greater than typical PCR/sequencing errors. One drawback of the HiSeq 4000 platform is a relatively high probability of index hopping. However, index hopping is unlikely in our experiments for the following reasons. First, we used two PCRs (Fig. S1b): the first PCR added sample indices while the second PCR added Illumina adapters. The mixing of amplicons from different samples took place after two steps of PCR right before sequencing. The free adapters in the sequencing pool would not have both adapters and index sequences. Second, we performed PCR purification after each PCR, which should have removed the remaining primers.

## Sequencing-based expression level estimation

For the promoter-barcode pair sequencing, we first filtered the sequencing reads to ensure that the barcodes were 20-nucleotide long and promoters were 120-nucleotide long. We clustered promoters and barcodes separately. We used Bartender[45] to cluster the barcodes with a tolerance of two mismatches, and used CD-HIT[46] to cluster the promoters with a tolerance of six mismatches. We allowed relatively high numbers of mismatches because the promoters (or barcodes) are completely random such that two distinct promoters (or barcodes) have an exceedingly low probability to be different by fewer than 7 (or 3) nucleotides. We discarded any barcode cluster connected with different promoter clusters. For any promoter cluster connected with multiple barcode clusters, all of these barcodes were counted toward the promoter cluster.

For cDNA and DNA sequencing of the barcodes, we counted only one of the reads when these reads shared the same UMI. Afterwards, we divided the read count of each barcode by the total read count in each sample to obtain the normalized read count (NRC) for the barcode. The expression level of barcode $i$ is measured by $NRC_i^{cDNA}/NRC_i^{DNA}$.

## Promoter GC content and activity

Given that the GC content is positively correlated with nucleosome occupancy[41] and that nucleosomes are depleted in yeast promoters[47], random promoters with higher GC contents are expected to have lower activities. However, a trend opposite to this expectation was found. This trend cannot arise from the potential impact of the GC content on the efficiency of expression measurement because it was

the barcode not the promoter sequence that was amplified in expression quantification.

## TFBS identification and analysis

Based on the position weight matrices (PWMs) of 196 yeast TFs and the suggested cutoffs in the ScerTF database[37], we identified the TFBSs of all these TFs on the forward and reverse strands of the random promoters, respectively. The forward strand is the strand with the same orientation as the barcode transcription that we intended to measure. For each TF and each strand, we separated promoters into two groups based on whether there is at least one binding site in the promoter sequence. We then tested whether the promoter activity differs between the two groups by a Wilcoxon rank-sum test.

## Promoter sequences of *S. cerevisiae* native genes

The promoter sequences of native genes were defined by from −500 nucleotides to either the translation start site (Fig. S8) or the transcription start site (Fig. S9).

## Expressions of yeast intergenic regions

We analyzed the RNA-seq data from the YPD medium generated by Chou et al. [38]. Specifically, the data from 10 wild-type samples were used. The RNA-seq data from the SCD medium were generated by Berg et al. [39], and only the data from three wild-type samples were used.

The reference genome of strain S288C was dissected into genic and intergenic regions. Our genic regions are defined conventionally[48] and include both protein-coding genes and RNA genes (rRNAs, tRNAs, snRNAs, snoRNAs, and ncRNAs) based on the annotations from Ensembl (http://useast.ensembl.org/Saccharomyces_cerevisiae/Info/Index) and SGD (http://sgd-archive.yeastgenome.org/sequence/S288C_reference/rna/). Intergenic regions are the entire genome subtracting protein-coding genes with their UTRs, RNA genes, centromeres, telomeres, long terminal repeats (LTRs), and LTR retrotransposons from the reference genome. We combined the annotated UTRs from two previous studies[49,50] and made additional 200-nucleotide outward extensions of both 5′ and 3′ UTRs. For protein-coding genes without annotated UTRs, we allocated 200 nucleotides outward from each end of the coding sequence as 5′ and 3′ UTRs, respectively. For RNA genes, we also added 50 nucleotides to each end of the gene. By doing the above, we aimed to minimize the influence of transcription from nearby genes or special sequence features on intergenic expression measures. We also varied the UTR extension length but found the results similar (Fig. S14e).

To measure intergenic expressions, we used a sliding window approach. For each intergenic region, we divided it into sliding windows of 20 nucleotides with a step size of 20 nucleotides, starting from the site of the intergenic region with the smaller genomic coordinate. We discarded the window at the end of an intergenic region if it is shorter than 20 nucleotides. The expression level of a window is measured by the number of reads mapped to the window in a strand-specific manner normalized by the total number of reads (in millions) of the RNA-seq data. For genic regions, a similar approach was used. For all windows from the same gene, their expressions were averaged to obtain the expression estimate for the gene. To estimate genic expressions by a canonical method, we employed the software StringTie[51].

## Comparing the expressions of intergenic windows or barcodes with the median expression level of yeast native genes

We divided the expression level of each intergenic window by the median expression level of all annotated genes (including RNA genes) in *S. cerevisiae* to obtain the relative expression level (*REL*) in each replicate. We then used a one-sample *t*-test to test if the *REL* is significantly different from 1 at a false discovery rate (FDR) of 0.05 by Benjamini−Hochberg's procedure[52]. We similarly tested if the *REL*

differs significantly from 0.1, 0.2, and so on. This allows computing the fraction of intergenic windows with *REL*s significantly higher than any *REL* cutoff (Fig. 4c), as well as identifying intergenic windows whose *REL*s are significantly higher than an *REL* cutoff but not significantly higher than the next (higher) cutoff (Fig. 4d).

For barcode *i*, we first merged its NRC from the three replicates to obtain the total NRC (TNRC) and estimated its expression level $E_i = TNRC_i^{cDNA}/TNRC_i^{DNA}$. Because eight of the barcodes are associated with the positive control, we also had eight $E_i$ values for the positive control. For each random promoter-associated barcode *i*, we obtained eight estimates of its expression relative to the positive control by $E_i/E_j$, where $j = 1$ to 8 refers to the eight barcodes of the positive control. Based on the expression level of *PSP2* relative to the median expression level of all yeast genes in RNA-seq data, we obtained the eight estimates of *REL* of each barcode, which is the expression level of the barcode relative to the median expression level of yeast genes. Finally, we used a one-sample *t*-test to test if the *REL* of a barcode is significantly different from 1 at FDR = 0.05 by Benjamini−Hochberg's procedure. We similarly tested if the *REL* of a barcode differs significantly from 0.1, 0.2, and so on. This allows computing the fraction of barcodes with *REL*s significantly higher than any *REL* cutoff (Fig. 4c), as well as identifying barcodes whose *REL*s are significantly higher than an *REL* cutoff but not significantly higher than the next (higher) cutoff (Fig. 4d). We similarly used a one-sample *t*-test to test if the expression level of barcode *i* ($E_i$) is significantly different from the negative control (using the eight expression estimates of the negative control) at FDR = 0.05 by Benjamini−Hochberg's procedure.

## Environment-specific expressions

First, expression levels of barcodes or intergenic windows are measured relative to the reference, which is the median expression level of yeast native genes. Second, at each expression level cutoff (0.1, 0.2, 0.3, …), we calculated the fraction of barcodes or intergenic windows with significantly higher expressions than the cutoff in YPD and SCD, respectively. Third, for barcodes or intergenic windows with significantly higher expressions than the cutoff in one environment (YPD or SCD), we calculated the fraction of them with significantly higher expressions than the same cutoff in the other environment; the fraction is referred to as the shared fraction.

## Correlation between the expression of an intergenic region and that of their neighboring genes

For each intergenic region, we considered the transcriptions of the two directions separately. For a given transcriptional direction, the mean expression of all windows in an intergenic region was used as an estimate of the expression level of the intergenic region for that direction. To consider its neighboring gene expression, we used (1) the mean expression level of both of its neighboring genes (one on each side), (2) the expression level of the upstream neighboring gene given the transcriptional direction under consideration, or (3) the readthrough level of the upstream neighbor, which is the expression level of the upstream neighboring gene only when it has the same transcriptional direction as the direction under consideration; otherwise, the expression of the neighbor is set at 0. We then correlated the expression level of an intergenic region with the expression of its neighboring genes in each of the above three ways.

## Intergenic transcriptions unattributable to neighboring gene expressions

The highest *REL* bin where expressions of intergenic windows are fully attributable to chance promoter activities is 0.3–0.4 in YPD (Fig. 4d) and 0.0–0.1 in SCD (Fig. S15b), respectively. For intergenic windows with *REL* > 0.4 in YPD (or >0.1 in SCD), we ranked them based on their neighboring gene expression level (following the first definition of neighboring gene expression in the preceding section) and then

separated them into 50 groups with equal numbers of intergenic windows per group according to the ranking. We iteratively removed the group with the highest neighboring gene expression. For the remaining groups, we calculated the median expression level of their neighboring genes ($ME_1$). From all intergenic windows (regardless of $REL$), we sampled the same number of windows as that in the remaining groups and calculated the median expression level of their neighboring genes ($ME_2$); this sampling was repeated 1000 times to allow the estimation of the fraction of times when $ME_2 \geq ME_1$. When this fraction exceeds the cutoff of 0.05, the remaining intergenic windows should be minimally influenced by the expressions of neighboring genes. We varied the cutoff and found the result similar (Fig. S17).

**Fraction of intergenic regions whose expressions are explainable by chance promoter activities**

For each intergenic region, we used the maximal (or the 95th percentile) expression level of its windows as a proxy for its expression level $E_{obs}$. Given the length ($L$) of the intergenic region, we calculated the number of nonoverlapping 120-nucleotide segments within the intergenic region by $N = \left[\frac{L}{120}\right]$, where $[x]$ is the smallest integer equal to or greater than $x$. We then randomly chose $N$ promoters from the random promoter library and compared the maximal expression of them ($E_{sampled}$) with $E_{obs}$. We repeated the random sampling 100,000 times and used the fraction of times when $E_{sampled} > E_{obs}$ as the nominal $P$ value for the null hypothesis that the expression of an intergenic region can be explained by chance promoter activities. The number of intergenic regions with expressions that cannot be explained by chance promoter activities is calculated by counting the number of intergenic regions with a $P$ value <0.05 upon a Benjamini–Hochberg multiple-testing correction. The expressions of neighboring genes of the intergenic regions whose expressions cannot be explained by chance promoter activities are not significantly higher than the corresponding values for the rest of the intergenic regions (Table S4). Hence, there is no need to correct the potential influence of neighboring gene expressions on intergenic expressions in this analysis.

**Reporting summary**

Further information on research design is available in the Nature Portfolio Reporting Summary linked to this article.

## Data availability

Sequencing data have been submitted to NCBI under accession code PRJNA876017. The RNA-seq datasets used for intergenic expression analysis are available under the accession number PRJNA728585 and PRJNA392312. Additionally, data files for active promoters are available at https://github.com/JasperXuEvolution/Random_promoter/tree/main/Data. Intermediate data files are available at https://figshare.com/articles/dataset/Intermediate_data_for_Chance_promoter_activities_illuminate_the_origins_of_eukaryotic_intergenic_transcriptions_/22231603. Source data are provided with this paper.

## Code availability

Computer code is available at https://github.com/JasperXuEvolution/Random_promoter.

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

## Acknowledgements

We thank D. Jiang, A. Mahilkar, and X. Shen for valuable comments. This work was supported by U.S. National Institutes of Health research grant R35GM139484 to J.Z.

## Author contributions

J.Z. conceived of the project and acquired funding; H.X., C.L., and J.Z. designed the study; C.L. performed some pilot experiments; H.X. performed the experiments; H.X. and C.X. analyzed the data; H.X. and J.Z. wrote the paper, with input from all authors.

## Competing interests

The authors declare no competing interests.
