## [Peer Review File · Nature Communications]

Chance promoter activities illuminate the origins of eukaryotic intergenic transcriptionsReviewers' Comments:

Reviewer #1:

Remarks to the Author:

The study by Xu et al addresses the question of whether the extensive transcription from intergenic regions across the genome is likely to be functional. They attempt to establish an empirical null distribution of intergenic activities by measuring the ability of random sequences to promote transcription from a common location in the yeast genome. By comparing this distribution to the distribution of activities from native yeast intergenic regions the authors conclude that the vast majority of intergenic transcription in yeast is consistent with the non-functional hypothesis. As presented, it is not clear whether the data support this conclusion.

The main claim of this paper is that 35-60% of random sequences drive transcription over the background levels. To estimate this fraction, the authors compute the activity of each random promoter by summing the cDNA and DNA read counts over all the replicates and calculating one ratio. They then use a one-sample t-test to compare this computed activity to the activity of the positive and negative controls. My main concern with this approach is that the variability of the replicates is lost by averaging over all the replicates. This may inflate the false-positive rate and give an incorrect estimate of the proportion of random promoters with activity above the background level. What is the justification for using the one-sample test instead of the more standard two-sample test that accounts for the variability among replicates, which the authors show in Figure 1 to be high for a large fraction of the random promoters?

The second major claim of the paper is that most intergenic transcription in yeast can be attributed to chance promoter activity. By comparing the distribution of activities from random sequences to bulk RNA sequencing datasets from prior studies the authors estimate the fraction of activities that are explainable by chance. What is the justification for directly comparing counts of reads mapping to barcodes (measured by primers specific to the barcode) and counts of RNA-seq reads from intergenic reads? At first pass these quantities do not appear to be directly comparable. The authors use the median of all native genes' expression as a reference to normalize these two quantities, however it is unclear how the barcode counts can be normalized using the median expression of a gene from RNA-seq as these two types of measurements are made using different molecular biology steps and are on different scales. This issue confounds the estimate of the fraction of intergenic windows that can be ascribed to chance promoter activity.

Reviewer #2:

Remarks to the Author:

The findings in this manuscript constitute an important contribution to the controversy of whether every detected transcript in a cell has a function or not (PMID: 25081515, PMID: 26818079). Common sense would dictate that a large proportion of extra DNA and random transcripts in most Eukaryotes is not functional but is an essential playground or cauldron for evolution (PMID: 14500911; page 16 in PMID: 16093654). This manuscript provides important experimental proof. Furthermore, the data are of relevance for those, who are interested in structure/function relationships of transcription promoters in Eukaryotes, specifically in yeast. For that reason, it would be useful to provide the sequences of at least the strongest promoters in a supplementary figure along with annotation of promoter elements, such as TATA box or TF binding-sites.

It would be very useful to have a more detailed description of the locus of integration, especially within the context of the inserted sequence and the resulting transcripts. For example, how long is the expected transcript, does it terminate before it reaches the downstream gene? Is there a terminator also downstream, which would be useful to obtain a defined transcript that is identical for all constructs at least with respect to the 3' moiety? The 5' ends are expected to vary, depending on the

placement of the transcription start site and on the sequence of the random barcode. Does the 3' moiety have a potential secondary structure? These considerations are important – at least under quantitative aspects, because the data do not purely reflect promoter strength, but can be influenced significantly by stabilities of the various composite transcript

Other points:

Line 52: perhaps a sentence is needed to define "genic regions" pertaining to this manuscript. Does it, in addition to promoters, exons, introns, and terminators, also include enhancer regions? Does it include RNA coding genes and exclude non-functional but annotated transcripts? The reference by Tsai et al. (2017) "Defining Functional Genic Regions in the Human Genome through Integration of Biochemical, Evolutionary, and Genetic Evidence", PMID 28398576 could be cited for definition, if it matches the authors'.

In the introduction, intronic transcription should be mentioned for comprehensiveness, although it is not too relevant for *S. cerevisiae* due to its paucity of introns. In Eukaryotes, many functional RNAs, such as snoRNAs are encoded in introns. Also, it is well known and should be mentioned that the act of transcription (transcriptional interference, promoter occlusion) is a means of regulating the activity of downstream promoters and generates transcripts without further function (Martens et al. (2004) "Intergenic transcription is required to repress the *Saccharomyces cerevisiae* SER3 gene", PMID: 15175754; see also PMID: 29309647; PMID: 32133533).

Line 254: non-sharing under different growth condition is only one indicator of possible functionality of a transcript, but is, by no means a guarantee.

Typos:

Line 275: "PCR errors are..."

Line 523: fraction instead of "faction"

Response to reviewers

We thank the two reviewers for their valuable comments, which have helped improve our manuscript. Below please find our point-to-point response in blue.

Reviewer #1

The study by Xu et al addresses the question of whether the extensive transcription from intergenic regions across the genome is likely to be functional. They attempt to establish an empirical null distribution of intergenic activities by measuring the ability of random sequences to promote transcription from a common location in the yeast genome. By comparing this distribution to the distribution of activities from native yeast intergenic regions the authors conclude that the vast majority of intergenic transcription in yeast is consistent with the non-functional hypothesis. As presented, it is not clear whether the data support this conclusion.

The main claim of this paper is that 35-60% of random sequences drive transcription over the background levels. To estimate this fraction, the authors compute the activity of each random promoter by summing the cDNA and DNA read counts over all the replicates and calculating one ratio. They then use a one-sample *t*-test to compare this computed activity to the activity of the positive and negative controls. My main concern with this approach is that the variability of the replicates is lost by averaging over all the replicates. This may inflate the false-positive rate and give an incorrect estimate of the proportion of random promoters with activity above the background level. What is the justification for using the one-sample test instead of the more standard two-sample test that accounts for the variability among replicates, which the authors show in Figure 1 to be high for a large fraction of the random promoters?

We agree that using a one-sample *t*-test with summed data from three replicates (*i.e.*, combined analysis) may inflate the false-positive rate. We did not use a two-sample *t*-test because the relatively small number of replicates renders the test powerless. To assess the reliability of our results from the combined analysis, we performed a thorough reanalysis using the data from each replicate individually (*i.e.*, individual analysis).

In YPD, the fraction of random promoters with significant activities is 63% in the combined analysis, and is 49%, 62%, and 51% in the three individual analyses, respectively (see the newly added Table S5). In SCD, the fraction of random promoters with significant activities is 41% in the combined analysis, and is 34%, 33%, and 34% in the three individual analyses, respectively (Table S5). Furthermore, the above results are generally robust when barcodes with at least 100 DNA reads are considered (see the newly added Fig. S20ab). Therefore, although variations among biological replicates exist, their effects are relatively minor and the conclusions from the combined analysis generally hold at least qualitatively.

A related question is whether the active promoters discovered from each individual analysis overlap with those discovered by the combined analysis. As shown in Fig. S20cd, regardless of the growth condition or sample, >74% of the active promoters of an individual analysis overlap with those from the combined analysis.

Furthermore, we re-estimated the fraction of random promoters with significantly higher promoter activities than the reference and the fraction of intergenic windows whose expressions are unexplained by chance promoter activities in YPD or SCD using individual analysis. The results are generally consistent with those from the combined analysis (Table S5).

We have added a section at the end of Results to describe these individual analyses and have added Fig. S20 and Table S5 to present the results obtained.

The second major claim of the paper is that most intergenic transcription in yeast can be attributed to chance promoter activity. By comparing the distribution of activities from random sequences to bulk RNA sequencing datasets from prior studies the authors estimate the fraction of activities that are explainable by chance. What is the justification for directly comparing counts of reads mapping to barcodes (measured by primers specific to the barcode) and counts of RNA-seq reads from intergenic reads? At first pass these quantities do not appear to be directly comparable. The authors use the median of all native genes' expression as a reference to normalize these two quantities, however it is unclear how the barcode counts can be normalized using the median expression of a gene from RNA-seq as these two types of measurements are made using different molecular biology steps and are on different scales. This issue confounds the estimate of the fraction of intergenic windows that can be ascribed to chance promoter activity.

As the reviewer pointed out, one cannot directly compare the number of reads mapped to barcodes and the number of RNA-seq reads mapped to intergenic regions. Therefore, we used a positive control, *PSP2* promoter, in our random promoter library as an internal control because the *PSP2* gene expression level is also available from the RNA-seq data. In other words, we measured a random promoter activity relative to the *PSP2* promoter activity. Because *PSP2* is an arbitrarily chosen gene, we further computed the expression level of *PSP2* relative to the median yeast gene expression level using the RNA-seq data. This way, we computed the expression level of each random promoter relative to the median gene expression level in yeast. This conversion is described in the main text (line 141-144) and Methods. Each intergenic expression level was also computed relative to the yeast median gene expression level. Hence, random promoter activities and intergenic expressions became comparable.

Reviewer #2

The findings in this manuscript constitute an important contribution to the controversy of whether every detected transcript in a cell has a function or not (PMID: 25081515, PMID: 26818079). Common sense would dictate that a large proportion of extra DNA and random transcripts in most Eukaryotes is not functional but is an essential playground or cauldron for evolution (PMID: 14500911; page 16 in PMID: 16093654). This manuscript provides important experimental proof. Furthermore, the data are of relevance for those, who are interested in structure/function relationships of transcription promoters in Eukaryotes, specifically in yeast. For that reason, it would be useful to provide the sequences of at least the strongest promoters in a supplementary figure along with annotation of promoter elements, such as TATA box or TF binding-sites.

We thank the reviewer for the overall positive evaluation. To address this comment, we added Fig. S13, which includes the strongest random promoter in YPD, strongest random promoter in SCD, and a strong promoter active in both conditions. Because one random promoter sequence will be bound by around 40 TFs on average when a certain level of mismatch is allowed, for illustration, we only label those perfectly matched transcription factor binding sites. We have uploaded two files to GitHub (https://github.com/JasperXuEvolution/Random_promoter) that show features of active random promoters in YPD and SCD, respectively. The features include the promoter sequence, promoter activity, GC content, names of TFs that potentially bind to the promoter, and the presence/absence of a TATA box.

It would be very useful to have a more detailed description of the locus of integration, especially within the context of the inserted sequence and the resulting transcripts. For example, how long is the expected transcript, does it terminate before it reaches the downstream gene? Is there a terminator also downstream, which would be useful to obtain a defined transcript that is identical for all constructs at least with respect to the 3' moiety? The 5' ends are expected to vary, depending on the placement of the transcription start site and on the sequence of the random barcode. Does the 3' moiety have a potential secondary structure? These considerations are important – at least under quantitative aspects, because the data do not purely reflect promoter strength, but can be influenced significantly by stabilities of the various composite transcript

The integration site is between gene *HSP31* and *FIT1* on Chr IV (mentioned on line 314 of the manuscript) and the coordinate is shown in Table S1. It is an intergenic site with few reads mapped in prior RNA-seq data. The detailed structure of our integration cassette is in Fig. S1.

Because we measured the expression level using the 20-nucleotide random barcode, which is just downstream of the random promoter, we did not study where the transcripts start/terminate and do not know the lengths of the transcripts. We put a terminator upstream but not downstream of our construct, because we were more concerned with transcriptions from the upstream region that may interfere with our estimation of promoter strengths. Because different transcripts have the same corresponding DNA sequences downstream of the 20-nucleotide barcode region, the comparison among their measured expression levels should be fair. This comparison is at least as good as the expression level comparison among native genes by RNA-seq, because different native genes have different sequences.

Other points:

Line 52: perhaps a sentence is needed to define “genic regions” pertaining to this manuscript. Does it, in addition to promoters, exons, introns, and terminators, also include enhancer regions? Does it include RNA coding genes and exclude non-functional but annotated transcripts? The reference by Tsai et al. (2017) “Defining Functional Genic Regions in the Human Genome through Integration of Biochemical, Evolutionary, and Genetic Evidence”, PMID 28398576 could be cited for definition, if it matches the authors’.

Our “genic region” contains protein-coding genes with their UTRs, RNA genes, and annotated transcripts. A detailed explanation of our definition of the intergenic region can be found in

Methods (the paragraph starting line 459). In yeast, there are upstream activation sequences (UASs), similar to enhancers in humans. The UASs are usually very close to the translation start site. Since we extend UTR outward 200 bp in each direction, the UASs are likely included in the genic region. More importantly, we varied the extension of UTR up to 1.6 kb, and the results are similar (Fig. S14e). Our definition of the genic region is similar to the reference suggested by the reviewer, so we have added this reference.

In the introduction, intronic transcription should be mentioned for comprehensiveness, although it is not too relevant for *S. cerevisiae* due to its paucity of introns. In Eukaryotes, many functional RNAs, such as snoRNAs are encoded in introns. Also, it is well known and should be mentioned that the act of transcription (transcriptional interference, promoter occlusion) is a means of regulating the activity of downstream promoters and generates transcripts without further function (Martens et al. (2004) “Intergenic transcription is required to repress the *Saccharomyces cerevisiae* SER3 gene”, PMID: 15175754; see also PMID: 29309647; PMID: 32133533).

We have now mentioned in Introduction the possibility that the act of transcription *per se* is functional and added the references suggested. As part of genes, introns are naturally transcribed. We agree that introns may contain functional elements, but because our study does not attempt to explain intronic transcription, we decide to focus on intergenic transcription in Introduction to avoid unnecessary confusion.

Line 254: non-sharing under different growth condition is only one indicator of possible functionality of a transcript, but is, by no means a guarantee.

We agree and have modified the statement.

Typos:

Line 275: “PCR errors are...”

Line 523: fraction instead of “faction”

We have fixed the typos.

Reviewers' Comments:

Reviewer #1:

Remarks to the Author:

The authors have satisfactorily addressed the two concerns I raised. The results from individual analyses of replicate experiments do generally agree with the analysis of the pooled data. I also appreciate the clarification for how the normalization of barcode counts to RNA-seq was performed. I think the manuscript is an important data point in the ongoing dialectic about the functionality of pervasive intergenic transcription.

Reviewer #2:

Remarks to the Author:

The authors have reasonably addressed my previous suggestions.

Below are some very minor points:

Fig. S13 Title In the figure itself "A strong random promoter in both conditions" should be "A strong random promoter in both conditions"

I am myself not sure, but in the abstract some of the present tense occurrences should be past tense? Perhaps the editors could advise?

"We approach this question by...

We build a library of over...

Quantifying the RNA concentration of each barcode in two environments reveals that 41-63% of random sequences have significant, albeit...

We find that only 1-5% of yeast intergenic transcriptions are unattributable to chance promoter activities or neighboring gene expressions, and these transcriptions exhibit higher-than-expected environment-specificity.

These findings suggest that only a minute fraction of intergenic transcription is functional in yeast."

We approached this question by...

"We built a library of over...

Quantifying the RNA concentration of each barcode in two environments revealed that 41-63% of random sequences had significant, albeit...

We find that only 1-5% of yeast intergenic transcriptions were unattributable to chance promoter activities or neighboring gene expressions, and these transcriptions exhibited higher-than-expected environment-specificity.

These findings suggested that only a minute fraction of intergenic transcription is functional in yeast."

Response to reviewers

We thank the two reviewers for re-reviewing our manuscript. Below please find our point-to-point response in blue.

Reviewer #1

The authors have satisfactorily addressed the two concerns I raised. The results from individual analyses of replicate experiments do generally agree with the analysis of the pooled data. I also appreciate the clarification for how the normalization of barcode counts to RNA-seq was performed. I think the manuscript is an important data point in the ongoing dialectic about the functionality of pervasive intergenic transcription.

We thank the reviewer for the positive evaluation.

Reviewer #2

The authors have reasonably addressed my previous suggestions.

Below are some very minor points:

Fig. S13 Title In the figure itself “A strong random promoter in both conditions” should be “A strong random promoter in both conditions”

The typo in Fig. S13 has been corrected.

I am myself not sure, but in the abstract some of the present tense occurrences should be past tense? Perhaps the editors could advise?

“We approach this question by...

We build a library of over...

Quantifying the RNA concentration of each barcode in two environments reveals that 41-63% of random sequences have significant, albeit...

We find that only 1-5% of yeast intergenic transcriptions are unattributable to chance promoter activities or neighboring gene expressions, and these transcriptions exhibit higher-than-expected environment-specificity.

These findings suggest that only a minute fraction of intergenic transcription is functional in yeast.”

We approached this question by...

“We built a library of over...

Quantifying the RNA concentration of each barcode in two environments revealed that 41-63% of random sequences had significant, albeit...

We find that only 1-5% of yeast intergenic transcriptions were unattributable to chance promoter activities or neighboring gene expressions, and these transcriptions exhibited higher-than-expected environment-specificity.

These findings suggested that only a minute fraction of intergenic transcription is functional in yeast.”

The editor confirmed that using the present tense is appropriate.